# Trace Amine-Associated Receptor 1 (TAAR1) Is a Positive Prognosticator for Epithelial Ovarian Cancer

**DOI:** 10.3390/ijms22168479

**Published:** 2021-08-06

**Authors:** Tilman L. R. Vogelsang, Aurelia Vattai, Elisa Schmoeckel, Till Kaltofen, Anca Chelariu-Raicu, Mingjun Zheng, Sven Mahner, Doris Mayr, Udo Jeschke, Fabian Trillsch

**Affiliations:** 1Department of Obstetrics and Gynecology, University Hospital, LMU Munich, 80337 Munich, Germany; ti.vogelsang@campus.lmu.de (T.L.R.V.); Aurelia.Vattai@med.uni-muenchen.de (A.V.); till.kaltofen@med.uni-muenchen.de (T.K.); Anca.ChelariuRaicu@med.uni-muenchen.de (A.C.-R.); Mingjun.Zheng@med.uni-muenchen.de (M.Z.); sven.mahner@med.uni-muenchen.de (S.M.); fabian.trillsch@med.uni-muenchen.de (F.T.); 2Faculty of Medicine, Institute of Pathology, LMU Munich, 80337 Munich, Germany; elisa.schmoeckel@med.uni-muenchen.de (E.S.); Doris.Mayr@med.uni-muenchen.de (D.M.); 3Department of Obstetrics and Gynecology, University Hospital Augsburg, 86156 Augsburg, Germany

**Keywords:** TAAR1, ovarian cancer, prognostic factor, immunohistochemistry, overall survival, 3-iodothyronamine

## Abstract

Trace amine-associated receptor 1 (TAAR1) is a G_αs_- protein coupled receptor that plays an important role in the regulation of the immune system and neurotransmission in the CNS. In ovarian cancer cell lines, stimulation of TAAR1 via 3-iodothyronamine (T_1_AM) reduces cell viability and induces cell death and DNA damage. Aim of this study was to evaluate the prognostic value of TAAR1 on overall survival of ovarian carcinoma patients and the correlation of TAAR1 expression with clinical parameters. Ovarian cancer tissue of *n* = 156 patients who were diagnosed with epithelial ovarian cancer (serous, *n* = 110 (high-grade, *n* = 80; low-grade, *n* = 24; unknown, *n* = 6); clear cell, *n* = 12; endometrioid, *n* = 21; mucinous, *n* = 13), and who underwent surgery at the Department of Obstetrics and Gynecology, University Hospital of the Ludwig-Maximilians University Munich, Germany between 1990 and 2002, were analyzed. The tissue was stained immunohistochemically with anti-TAAR1 and evaluated with the semiquantitative immunoreactive score (IRS). TAAR1 expression was correlated with grading, FIGO and TNM-classification, and analyzed via the Spearman’s rank correlation coefficient. Further statistical analysis was obtained using nonparametric Kruskal-Wallis rank-sum test and Mann-Whitney-U-test. This study shows that high TAAR1 expression is a positive prognosticator for overall survival in ovarian cancer patients and is significantly enhanced in low-grade serous carcinomas compared to high-grade serous carcinomas. The influence of TAAR1 as a positive prognosticator on overall survival indicates a potential prognostic relevance of signal transduction of thyroid hormone derivatives in epithelial ovarian cancer. Further studies are required to evaluate TAAR1 and its role in the development of ovarian cancer.

## 1. Introduction

With more than 7000 new cases in Germany in 2016, ovarian cancer (OC) is representing 3.1% of cancer cases and 5.2% of cancer deaths in females [1]. Due to its high potential for cancer dissemination into the peritoneum, frequent late symptoms and detection, heterogeneity, and acquired and intrinsic chemoresistance [2,3], epithelial ovarian cancer shows relative 5 years survival rates of less than 50% [4] with only slight improvement and a wide range of 5 years survival rates in geographic distribution [5]. OC presents in a histologically diverse way, with epithelial type being the most frequent, accounting for more than 90% of all primary ovarian tumors [6]. Other histological subtypes include sex cord-stromal (5–6%), germ cell (2–3%), and rare miscellaneous entities [6]. This article focuses on epithelial tumors, which can be subdivided into high-grade serous carcinomas (70–80% of cases), low-grade serous carcinomas, endometrioid (<5%), clear cell (3%), and mucinous cancers (<3%) [7].

Therapeutic management of OC lacks effective screening methods. Clinical prognosticators such as disease stage at diagnosis, postsurgical residual disease, histological subtype, grading, mutation in the breast cancer gene (BRCA), general state of health, age, guideline-based therapy, and volume of ascites are of value [1,3]. Platinum resistance after chemotherapy can worsen the prognosis of ovarian cancer patients [8,9]. Attempts of establishing new prognostic factors have been published in multiple studies [10,11,12].

Ovarian cancer is regulated by thyroid hormones and its derivatives [13,14]. Thyroid hormones also have a pro-angiogenic role in various cancer types, making its receptors potential targets for therapeutic treatment [15]. Similar to ovarian cancer, hypothyroidism occurs predominantly in aging women [16,17]. Hyperthyroidism is prevalent in women from the 3rd to 5th decade of life with an 8:1 female:male ratio [18].

Thyronamines, such as 3-iodothyronamine (T_1_AM), derive from thyroid hormones [19] via degradation by the enzyme ornithine decarboxylase (ODC) and intestinal deiodinases [20]. In comparison with the mostly epigenetically acting classical thyroid hormones, decarboxylated thyroid hormones can function rapidly (e.g., via fast change of heart rate and body temperature) [19]. T_1_AM is described to inhibit cell growth and viability, induce cell death, and lead to DNA damage in ovarian cancer cells [13]. Furthermore, T_1_AM and its metabolites, T_0_AM and thyroacetic acid (TA_1_), can influence pancreatic islets, brain, heart, and other tissues through the G_αs_-protein coupled receptor trace amine-associated receptor 1 (TAAR1) [20]. Other agonists of TAAR1 are trace amines, which are closely associated metabolically with the dopamine, serotonin, and noradrenaline systems [21]. The degradation and synthesis of both trace amines and thyronamines proceed via enzymes working through decarboxylation. L-thyroxin (T4) is degraded to thyronamines (T_1_AM) via ornithine decarboxylase (ODC) [20] while trace amines are synthesized via L-amino acid decarboxylase [21]. TAAR1 is widely expressed including placenta, brain, spinal cord, immune cells such as leukocytes, macrophages and dendritic cells, breast cancer tissue, D-cells in stomach, and pancreatic β cells [22,23,24,25,26]. Activation of TAAR1 leads to a G_αS_-protein mediated increase in intracellular cAMP levels [27,28].

Our previously published study showed that an increased TAAR1 expression is correlated significantly with a positive survival rate in breast cancer patients [24]. To extend upon this finding, we aimed to analyze the expression of TAAR1 in another gynecological tumor entity. In detail, TAAR1 expression in ovarian epithelial cancer tissue was evaluated, and correlation with overall survival and clinical parameters was performed.

## 2. Results

### 2.1. Differences of TAAR1 Expression in Histological Subtypes of Ovarian Cancer

TAAR1 was detected in membrane as well as in cytoplasm. Membrane TAAR1 staining could be gained in *n* = 134 cases and presented in endometrioid tumors (*n* = 16) and serous carcinoma (*n* = 98), with a median IRS of 3, in clear cell (*n* = 9) and mucinous tumor (*n* = 11) with a median IRS of 1 (*p* = 0.003), as shown in Figure 1. In *n* = 22 cases, either staining was not successful, or no ovarian cancer tissue was hit.

Likewise, correlation of cytoplasmic TAAR1 expression and histological subtype differs significantly. A total of *n* = 128 cases could be observed. While endometrioid tumor (*n* = 19) showed a median IRS of 4, clear cell carcinoma (*n* = 10) showed a median IRS of 3.5, serous carcinoma a median IRS of 3 (*n* = 89), and mucinous carcinoma (*n* = 10) a median IRS of 2.5 (*p* = 0.009), as shown in Figure 2. On *n* = 28 slides, staining was not successful, or no ovarian cancer tissue was gained.

### 2.2. TAAR1 Expression in High-Grade and Low-Grade Serous Ovarian Cancer

*n* = 110 tissue sections from patients diagnosed with serous ovarian carcinoma were stained (high-grade, *n* = 80; low-grade, *n* = 24; unknown, *n* = 6). Due to missed hits of ovarian cancer tissue and/or failed staining in *n* = 17 cases, *n* = 93 slides were examined. Correlation of membrane TAAR1 expression, with grading of serous carcinoma, showed that TAAR1 is expressed significantly higher in low-grade serous carcinoma (median IRS of 4; *n* = 22) compared to high-grade serous carcinoma (median IRS of 3; *n* = 71) (*p* = 0.028) (Figure 3).

### 2.3. TAAR1 Expression Correlated with TNM Classification and FIGO

Correlation of TAAR1 expression and TNM classification was obtained. Ovarian clear cell carcinomas with smaller sizes of the primary tumor (pT1, *n* = 6) had a significantly higher TAAR1 expression (median IRS of 2.5) than clear cell carcinomas with a higher pT status (pT2, *n* = 3; pT3, *n* = 1) (median IRS of 0) (Figure 4a). Patients diagnosed with ovarian endometrioid carcinoma without local lymph node metastases (pN0, *n* = 7) had a significantly higher TAAR1 expression (median IRS of 4) than patients with local lymph node metastases (pN1, *n* = 3) (median IRS of 3) (Figure 4b).

There was no significant correlation of TAAR1 expression with distant metastases (pM).

### 2.4. Correlation of Membrane and Cytoplasmic TAAR1 Expression with Overall and Progression-Free Survival of Ovarian Cancer Patients

Ovarian cancer patients with enhanced TAAR1 expression in the membrane and/or cytoplasm (IRS > 3) (*n* = 69) have a significantly longer overall survival (OS) than patients with a low TAAR1 expression (IRS ≤ 3) (*n* = 64), as shown in the Kaplan-Meier curve in Figure 5 (*p* = 0.045). TAAR1 expression seems to be a positive prognostic factor for OS in ovarian cancer patients. To determine the best cut-off level for high and low TAAR1 expression, based on the maximal difference between specificity and sensitivity, a receiver operating characteristic curve (ROC-curve) was used. The cut-off level for high TAAR1 expression was IRS > 3.

Analysis of multiple variables, with Cox regression analysis for OS, showed that age, grading, and FIGO, but not TAAR1 expression, are independent prognosticators of OS (Table 1).

For progression-free survival, no significant difference but a tendency in TAAR1 expression could be detected. High cytoplasmic TAAR1 expression (IRS > 3) (*n* = 47) correlates with a worse progression-free survival (*p* = 0.105), whereas patients with low cytoplasmic TAAR1 expression (*n* = 86) have a better progression-free survival (Figure 6).

### 2.5. Correlations of TAAR1 and Other Variables

In previous studies, staining of the applied ovarian cancer collective with various markers had been obtained. Correlation analysis was performed using Spearman’s rank correlation coefficient. TAAR1 expression in the membrane significantly correlates with cytoplasmic TAAR1 expression (Spearman rho: 0.591, *p* = 0.000), estrogen receptor-α (ER-α) (Spearman rho: 0.342, *p* = 0.000), progesterone receptor-A (PR-A) (Spearman rho: 0.256, *p* = 0.004), progesterone receptor-B (PR-B) (Spearman rho: 0.250, *p* = 0.005) and negatively with nuclear vitamin D receptor (VDR) (Spearman rho: −0.178, *p* = 0.046) and G-protein-coupled estrogen receptor/G-protein-coupled receptor 30 (GPER/GPR30) (Spearman rho: −0.177, *p* = 0.042). Furthermore, membrane TAAR1 expression correlated with Muc-115D8 (Spearman rho: 0.226, *p* = 0.013), VU3C6 (Spearman rho: 0.180, *p* = 0.044) and negatively with Glycodelin A (Spearman rho: −0.223, *p* = 0.012) as shown in Appendix A. Muc-115D8 and VU3C6 are, similarly to HMFG1, VU4H5, and TA-MUC1, Mucin-1 (MUC1) epitopes generated by various glyco-modifications of MUC1 during the process of malignant transformation [29]. MUC1 is a high molecular weight transmembrane glycoprotein expressed at the luminal membrane of many types of physiological epithelial cells [30]. Disruption of the cell-cell and cell-matrix adhesions, due to MUC1 overexpression by the tumor cells, is believed to play a role in invasive cancer growth and metastasis [31,32,33,34,35]. Immunohistochemical detection of MUC1 via VU3C6 shows a correlation of MUC1 with pT but not with overall survival, grading, and FIGO in epithelial ovarian cancer patients [36]. Muc-115D8 expression in epithelial ovarian cancer patients is increased with advanced cancer stages [37].

Cytoplasmic TAAR1 expression correlates with ER-α (Spearman rho: 0.299, *p* = 0.000), PR-A (Spearman rho: 0.173, *p* = 0.049), PR-B (Spearman rho: 0.297, *p* = 0.001), and negatively with Glycodelin A (Spearman rho: −0.227, *p* = 0.009), as shown in Appendix A.

### 2.6. Correlations of TAAR1 Gene with Overall Survival and Progression Free Survival of Large Independent Ovarian Cancer Cohorts

To validate the overall survival and progression-free survival of the TAAR1 gene in large independent ovarian cancer patient cohorts, the *KM Plotter* database [38] was used. We divided patients into high and low groups based on the median expressions of TAAR1. Then, progression-free survival and overall survival were chosen to compare these groups. We selected to follow up the threshold for 240 months and exclude biased arrays that were selected for array quality control.

The results showed that the survival time of patients in the high-TAAR1 expression group (*n* = 170) was longer than that of the low-expression group (*n* = 485) for overall survival (Appendix A). While 8.2% of the patients in the high-TAAR1 expression group were alive after 100 months, only 7.0% were alive in the low-TAAR1 expression group after the same period. This difference was not significant.

The progression free survival of patients in the high-TAAR1 expression group (*n* = 299) was nearly identical with the progression free survival of patients in the low-expression group (*n* = 315) with no significant difference (Appendix A).

## 3. Discussion

In our study, we could show that ovarian cancer patients with high TAAR1 expression in the membrane and/or cytoplasm have a significantly longer overall survival (OS) compared to patients with a low TAAR1 expression and that high membrane TAAR1 expression correlates significantly with low-grade serous ovarian cancer.

Trace amine-associated receptor 1 (TAAR1) is a G_αs_-protein coupled receptor, which acts by the binding of thyronamines (e.g., 3-iodothyronamine (T_1_AM)), trace amines, and other agonists. This leads to the activation of TAAR1 and increasing cAMP levels [27,28]. TAAR1 can be expressed in membrane, cytoplasm, nucleus, and cytoskeleton [39]. According to our findings, TAAR1 expression is detectable in the cytoplasm and membrane of ovarian cancer tissue. In previously published studies, we could show that TAAR1 expression is a positive prognostic factor for OS in breast cancer patients [24] and that stimulation of breast cancer cell lines with T_1_AM leads to an increase in TAAR1 expression and a decrease in cell viability [40]. Likewise, the current study showed that general high TAAR1 expression is a positive prognostic factor for OS in ovarian cancer patients (*p* = 0.045). This finding is in line with a study of Shinderman-Maman and colleagues (2017), who showed that stimulation of ovarian cancer cell lines with T_1_AM leads to a decrease in cell viability and induces cell death and DNA damage [13]. Cox Regression analysis was carried out and showed that TAAR1 is not an independent factor for OS (Table 1). A reason might be that TAAR1 correlates significantly with multiple factors, such as histological subtype, grading, and partly TNM classification.

T_1_AM binds to TAAR1 and has neither an affinity to thyroid receptor (TR)-α or TR-β, nor the ability to modulate nuclear TR-mediated transactivation [41]. A widely known effect of the interaction between T_1_AM and TAAR1 is immediate reduction in body temperature and heart rate [19]. T_1_AM has the opposite effect to the thyroid hormones triiodothyronine (T_3_) and thyroxine (T_4_) (e.g., hyperthermia and increased cardiac output in hyperthyroidism) and acts more rapid compared to T_3_ and T_4_ [19,41]

Approximately 85–90% of ovarian malignancies derive from the ovarian surface epithelium (OSE) [42]. Stimulation of OSE cells with 17β-estradiol, which binds to ER-α and GPER/GPR30 [43], leads to oxidative DNA damages [44] and increases proliferation in ovarian surface epithelium [42]. Further in vivo studies show that treatment with 17β-estradiol decreases survival, in a transgenic mouse model, of ovarian cancer [45] and accelerates ovarian tumor progression in vivo [46]. In T_1_AM stimulated breast cancer cells, additional co-stimulation with estradiol leads to an upregulation of TAAR1 expression [40]. Interestingly, high ER-α expression in serous ovarian cancer is associated with longer OS [47] whereas the role of GPER/GPR30 in ovarian cancer patient survival is being discussed controversially [48,49,50]. In this study, we found that membrane, (Spearman rho: 0.342, *p* = 0.000) as well as cytoplasmic (Spearman rho: 0.299, *p* = 0.000), TAAR1 expression correlates significantly with ER-α expression in ovarian cancer tissue. Additionally, membrane TAAR1 expression correlates negatively with GPER/GPR30 (Spearman rho: −0.177, *p* = 0.042). Further in vitro studies are required to establish the exact interactions between TAAR1, ER-α and GPER/GPR30 in ovarian cancer.

A significant positive correlation between membranous TAAR1 expression and PR-A (Spearman rho: 0.256, *p* = 0.004) and PR-B (Spearman rho: 0.250, *p* = 0.005) could be found. Similarly, cytoplasmatic TAAR1 expression correlates significantly with PR-A (Spearman rho: 0.173, *p* = 0.049) and PR-B (Spearman rho: 0.297, *p* = 0.001). Progesterone receptors might have influence on the development of ovarian cancer and PR-B is known to be an independent positive prognostic factor for OS in ovarian cancer patients [51].

An association between TAAR1 and grading of ovarian serous carcinoma was found, showing that high membrane TAAR1 expression correlates with low-grade ovarian serous carcinoma (*p* = 0.028) (Figure 3). This is in line with the prognostic value of high TAAR1 expression and low-grade serous carcinoma, both of which are associated with better survival [52]. Furthermore, low grading (G1) in breast cancer tissue similarly correlates with high TAAR1 expression [24].

TAAR1 is well described in the neurological field where it can modulate the serotonergic and dopaminergic systems in brain tissue [53]. Dopamine receptor 2 (D2R) interacts with TAAR1 in the membrane of human embryonic kidney 293 (HEK-293) cells [54]. Stimulation with D2R-antagonists disrupts the interaction between D2R and TAAR1 and leads to a selectively enhanced TAAR1-mediated increase in cAMP [54]. Antagonism of D2R is described to have effects on tumorigenesis in ovarian cancer, cervical cancer, breast cancer, leukemia, and hepatoma [55,56,57,58,59]. Interestingly, some studies showed that dopamine can inhibit tumor growth and tumor angiogenesis in mouse ovarian tumor via its specific D2R [60,61]. In HEK-293 cells, TAAR1 is mainly expressed intracellularly, but membrane expression of TAAR1 increases with the presence of dopamine receptor D2 (D2R) [62]. The current study showed that high TAAR1 expression correlates significantly with better OS (*p* = 0.045) and thus, TAAR1 represents a positive prognosticator for ovarian cancer. It is of special interest to evaluate the exact D2R interaction with TAAR1 in ovarian cancer and its influence on tumorigenesis.

## 4. Materials and Methods

### 4.1. Patients

156 female patients who were diagnosed with ovarian cancer, and underwent surgery at the Department of Obstetrics and Gynecology, University Hospital of the Ludwig-Maximilians University Munich, Germany between 1990 and 2002, were included in this study. No patient received neoadjuvant chemotherapy. There was no preselection of patients. Munich Cancer Registry provided patient follow up data. The mean of patients’ age was 58.9 ± 12.5 years in the range of 30.3 to 88.0 years. In the course of the study, 104 deaths were observed with a mean OS of 3.2 ± 3.0 years.

Four histological subtypes of epithelial ovarian cancer were included into this study (serous (*n* = 110), clear cell (*n* = 12), endometrioid (*n* = 21), and mucinous (*n* = 13)) (Table 2). Tumor grading (G1 (*n* = 36), G2 (*n* = 11), G3 (*n* = 97)) was performed, according to WHO, by two experienced gynecological pathologists (E.S., D.M.). TNM classification (T = size or direct extent of the primary tumor, *n* = regional lymph node status, M = distant metastasis) was conducted according to the Union for International Cancer Control (UICC). Extent of the primary tumor (pT1 (*n* = 40), pT2 (*n* = 18), pT3 (*n* = 93), pT4 (*n* = 4)), local lymph node metastases (pN0 (*n* = 43), pN1 (*n* = 52)) and distant metastasis (pM0 (*n* = 3), pM1 (*n* = 6)) were evaluated (Table 2). FIGO stage was ascertained according to the criteria of the International Federation of Gynecology and Obstetrics.

### 4.2. Immunohistochemistry

TAAR1 expression was obtained in tissue micro arrays (TMAs), which were created from a collective of formalin-fixed, paraffin-embedded ovarian cancer samples. The experiment was initiated by deparaffinization of the slides (4 µm) by xylol. In order to inactivate endogenous peroxidase, 3% H_2_O_2_ in methanol (20 min) was used. Rehydration of the slides was obtained by a descending ethanol gradient. A pressure cooker filled with Epitope Retrieval Solution pH 9.0 (Novocastra by Leica, Wetzlar, Germany) provided the preparation of the tissue for heat induced epitope retrieval. Next, blocking solution was applied to prevent non-specific binding of the primary antibodies. The slides were incubated at room temperature for 1h with anti-TAAR1 (polyclonal rabbit IgG, Atlas Antibodies by BIOZOL, Eching, Germany) diluted 1:100. Antibody reactivity was detected using ImmPRESS Anti-Rabbit IgG Polymer Kit (Vector Lab. by BIOZOL, Eching, Germany) according to the manufacturer’s protocol, followed by application of substrate and chromogen (3,3′-diaminobenzidine Agilent Technologies, Waldbronn, Germany). In the next step, the slides were counterstained with hematoxylin Gill’s formula (Vector Lab.) and cover slipped.

### 4.3. Quantification

The slides were analyzed using a Leitz Diaplan microscope (Leitz, Wetzlar, Germany). Quantification of each slide’s staining was examined by application of the semiquantitative immunoreactive score (IRS). Therefore, optical estimation of intensity and distribution pattern of the antigen expression is obtained [63]. The IRS is calculated by multiplying staining intensity (0: none; 1: weak; 2: moderate; 3: strong) with the number of positively stained cells (in %) (0: no staining, 1: <10% of the cells; 2: 11–50%; 3: 51–80%; 4: >80%). A scale from 0 (no expression) to 12 (very high expression) was used. Provided photos were taken with a CCD color camera (JVC: Victor Company of Japan, Yokohama, Japan).

### 4.4. Statistics

IBM SPSS 26 Statistics for Windows, Version 26 (IBM Corp: Armonk, NY, USA), was used for data analysis. *p* values lower than *p* < 0.05 were considered statistically significant. Nonparametric Kruskal-Wallis rank-sum test and Mann-Whitney-U-test were used as appropriate for group comparisons regarding ordinal analysis variables. Correlations between variables were obtained using Spearman’s rank correlation coefficient. For survival analysis, we used Cox Mantel log rank test. Analysis of TAAR1 as an independent prognosticator was done using Cox regression analysis. IBM SPSS 25 Statistics for Windows, Version 26, as well as Microsoft^®^ PowerPoint for Mac Version 16.30 (19101301) were used for design of figures.

## 5. Conclusions

In this hypothesis generating study, we could observe that TAAR1 was a positive prognosticator for OS in ovarian cancer patients and was expressed significantly higher in low-grade serous carcinoma. The influence of TAAR1 on OS in epithelial ovarian cancer patients indicates a potential prognostic relevance of signal transduction of thyroid hormone derivates in ovarian cancer. Therefore, TAAR1 could be a novel therapeutic target in ovarian cancer patients. To establish the exact role of TAAR1 in ovarian cancer cells, further studies are required.

## Figures and Tables

**Figure 1 ijms-22-08479-f001:**
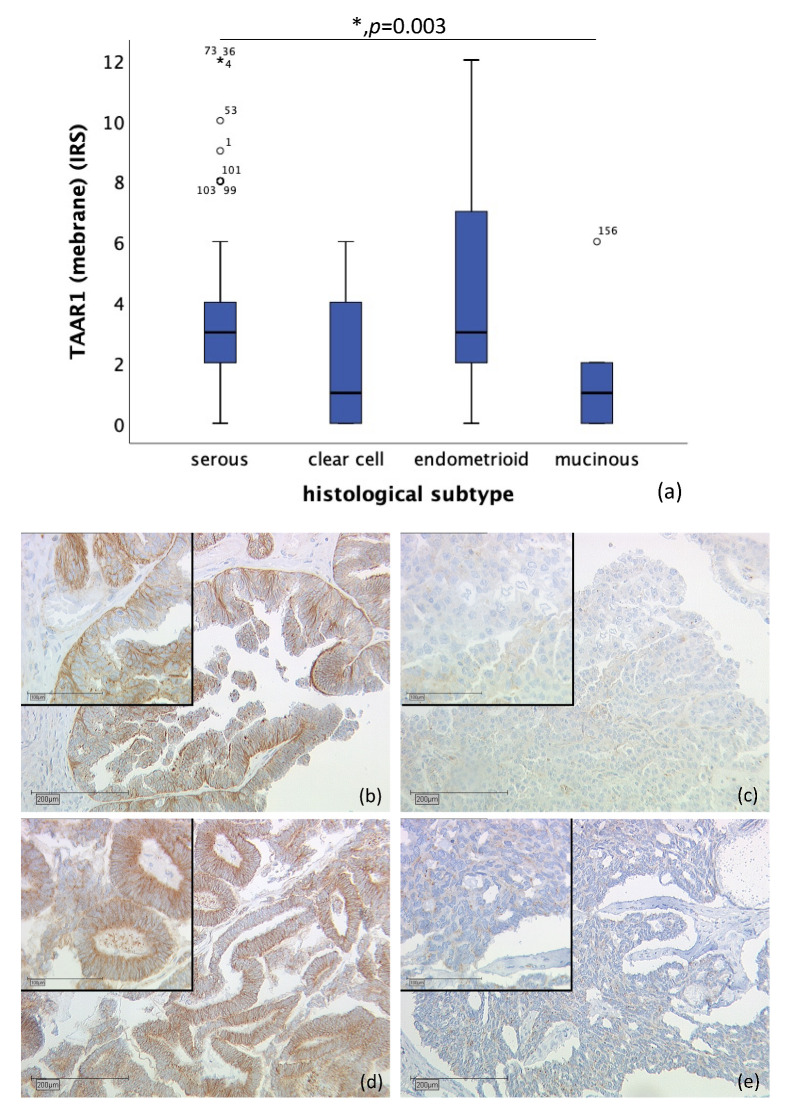
Correlation of membrane TAAR1 expression with histological subtype (*p* = 0.003) (**a**) boxplot of membrane TAAR1 expression and histological subtype (**b**) serous carcinoma (*n* = 98) with a membrane TAAR1 IRS of 4, magnification ×10 and ×25 in the inset (**c**) clear cell carcinoma (*n* = 9) with a membrane TAAR1 IRS of 1, magnification ×10 and ×25 in the inset (**d**) endometrioid carcinoma (*n* = 16) with a membrane TAAR1 IRS of 4, magnification ×10 and ×25 in the inset (**e**) mucinous carcinoma (*n* = 11) with a membrane TAAR1 IRS of 0, magnification ×10 and ×25 in the inset; * *p* < 0.05 was considered statistically significant.

**Figure 2 ijms-22-08479-f002:**
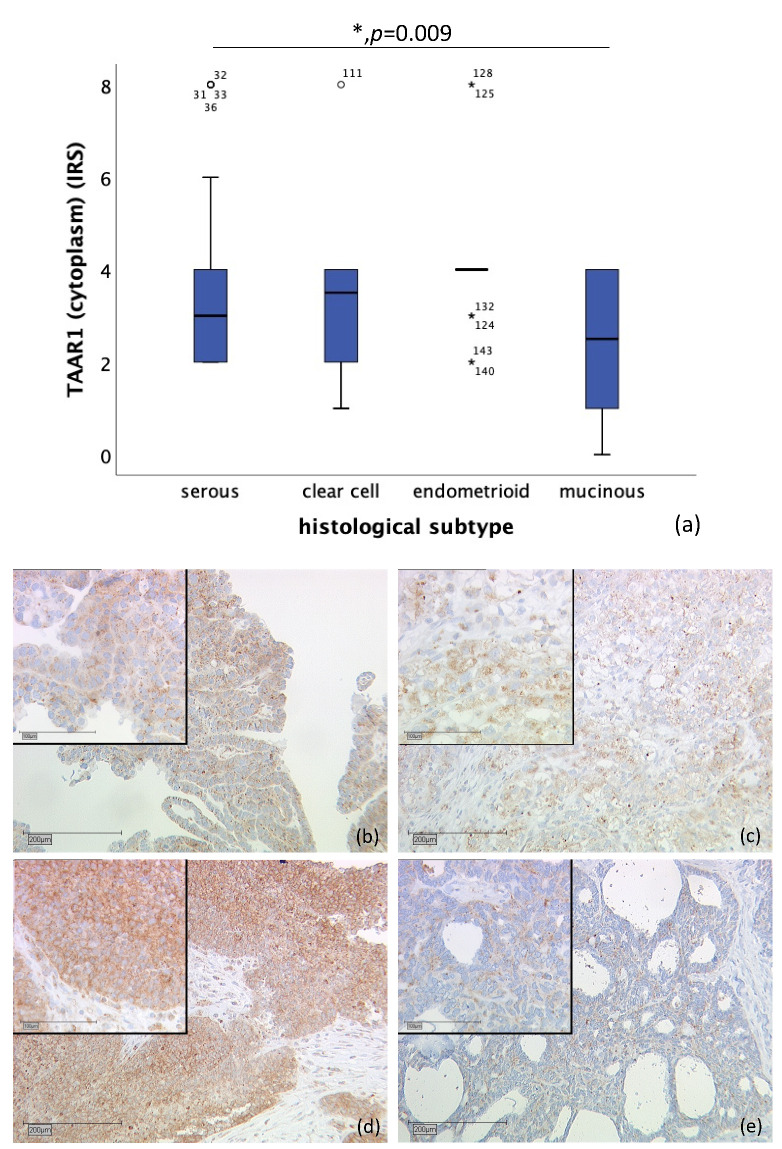
Correlation of cytoplasmic TAAR1 expression with histological subtype (*p* = 0.009) (**a**) boxplot of cytoplasmic TAAR1 expression and histological subtype (**b**) serous carcinoma (*n* = 89) with a cytoplasmic TAAR1 IRS of 3, magnification ×10 and ×25 in the inset (**c**) clear cell carcinoma (*n* = 10) with a cytoplasmic TAAR1 IRS of 3, magnification ×10 and ×25 in the inset (**d**) endometrioid carcinoma (*n* = 19) with a cytoplasmic TAAR1 IRS of 4, magnification ×10 and ×25 in the inset (**e**) mucinous carcinoma (*n* = 10) with a cytoplasmic TAAR1 IRS of 2, magnification ×10 and ×25 in the inset; * *p* < 0.05 was considered statistically significant.

**Figure 3 ijms-22-08479-f003:**
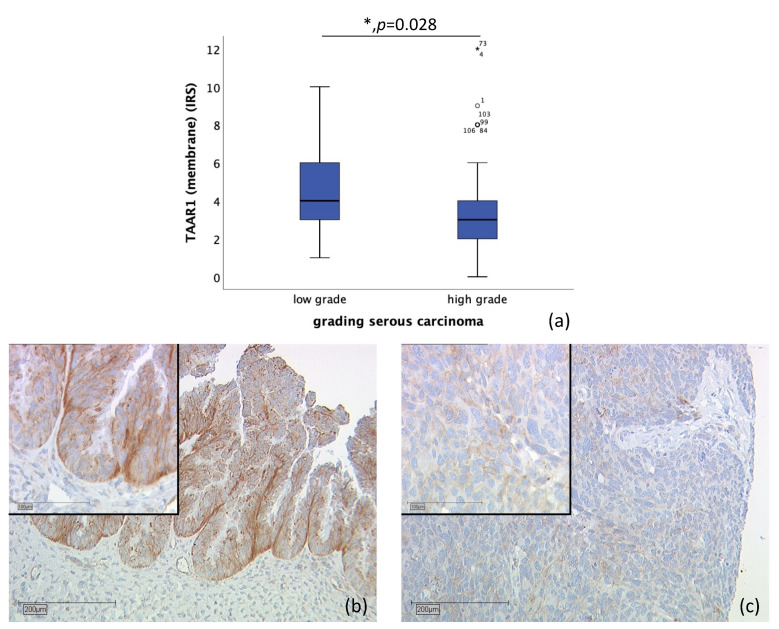
Correlation of membrane TAAR1 expression with grading in serous carcinoma (*p* = 0.028) (**a**) boxplot of membrane TAAR1 expression and grading in serous carcinoma (**b**) low grade serous carcinoma (*n* = 22) with a membrane TAAR1 IRS of 4, magnification ×10 and ×25 in the inset (**c**) high-grade serous carcinoma (*n* = 71) with a membrane TAAR1 IRS of 2, magnification ×10 and ×25 in the inset; * *p* < 0.05 was considered statistically significant.

**Figure 4 ijms-22-08479-f004:**
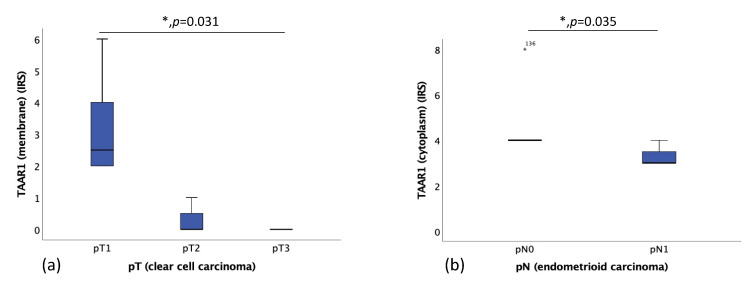
(**a**) Correlation of membrane TAAR1 expression with the size or extent of the primary tumor (pT) in clear cell carcinoma (*p* = 0.031), median IRS of pT1 is 2.5 (*n* = 6), median IRS of pT2 is 0 (*n* = 3), median IRS of pT3 is 0 (*n* = 1). (**b**) Correlation of membrane TAAR1 expression with local lymph node status (pN) in endometrioid carcinoma (*p* = 0.035), median IRS of pN0 is 4 (*n* = 7), median IRS of pN1 is 3 (*n* = 3); * *p* < 0.05 was considered statistically significant.

**Figure 5 ijms-22-08479-f005:**
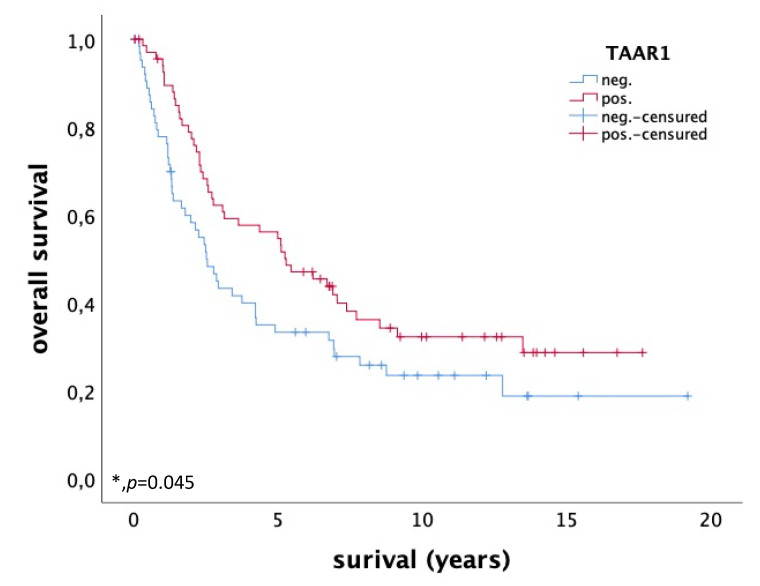
OS of patients with ovarian carcinoma with high (*n* = 69) and low (*n* = 64) TAAR1 expression. High TAAR1 expression is associated with a better OS in ovarian cancer patients (*p* = 0.045); * *p* < 0.05 was considered statistically significant.

**Figure 6 ijms-22-08479-f006:**
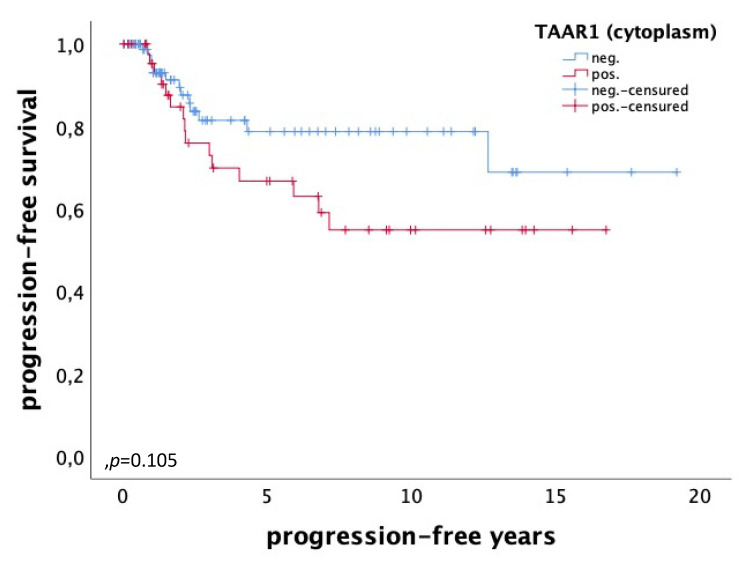
Progression-free survival of patients with ovarian carcinoma with a high (*n* = 47) and a low (*n* = 86) cytoplasmic TAAR1 expression. High cytoplasmic TAAR1 expression is tendentially associated with a worse progression-free survival in ovarian cancer patients (*p* = 0.105).

**Table 1 ijms-22-08479-t001:** Cox regression analysis.

Variable	Significance	Hazard Ratio	95% Confidence Interval	
			Lower	Higher
Age	0.040	1.022	1.001	1.044
Histological subtype	0.990	0.998	0.743	1.342
Grading	0.001	1.775	1.262	2.496
FIGO	0.001	1.963	1.323	2.913
TAAR1	0.344	0.807	0.518	1.258

**Table 2 ijms-22-08479-t002:** Patients’ characteristics.

	Number of Cases (Total Number of Cases: *n* = 156)	%
**Histopathological tumor subtype**		
Serous	110	70.5
Clear cell	12	7.7
Endometrioid	21	13.5
Mucinous	13	8.3
**Tumor grading**		
G1	36	23.1
G2	11	7.1
G3	97	62.2
Unknown	12	7.7
**Extent of primary tumor (pT)**		
pT1	40	25.6
pT2	18	11.5
pT3	93	59.6
pT4	4	2.6
Unknown	1	0.6
**Regional lymph node involvement (pN)**		
pN0	43	27.6
pN1	52	33.3
Unknown	61	39.1
**Presence of distant metastatic spread (pM)**		
pM0	3	1.9
pM1	6	3.9
Unknown	147	94.2
**FIGO classification**		
FIGO I	35	22.4
FIGO II	10	6.4
FIGO III	103	66.0
FIGO IV	3	1.9
Unknown	5	3.2

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
