# Peer review of "Trace Amine-Associated Receptor 1 (TAAR1) Is a Positive Prognosticator for Epithelial Ovarian Cancer"

_ijms, 2021, doi:10.3390/ijms22168479_

Round 1

Reviewer 1 Report

In this manuscript, the authors report that high TAAR1 expression may be an indicator of ovarian cancer survival, and show that it is significantly enhanced in low-grade serous carcinoma compared to high-grade serous carcinoma. A positive correlation was also suggested between TAAR1 expression and PR-A and PR-B.

The authors immunohistochemically stained cancer tissues with anti-TAAR1 based on clinical ovarian cancer samples and assessed them by semi-quantitative immune response scores (IRS).

However, this manuscript has weak validation of the role of TAAR1 in ovarian cancer tissue or cancer. Independent verification of this should be provided and these assumptions should be further investigated. In the mechanism part, how TAAR1 contributes to gene expression regulation is completely missing. Therefore, this manuscript is not accepted as is.

Based on clinical ovarian cancer samples, did the authors find a correlation between TAAR1 expression and disease state at the mRNA and protein level? The survival analysis may be studied using both mRNA and protein data such as from IHC data. Data from public database should be retrieved as well. Gene expression analysis requires performing genome-wide analysis of TAAR1-regulated genes to elucidate the most important functions of TAAR1 in ovarian cancer cells.

The study included 156 female patients diagnosed with ovarian cancer and who underwent surgery. Four histological subtypes of epithelial ovarian cancer were included in this study. Investigating TAAR1 expression in a relatively large population of patients with serous ovarian cancer and presenting functional findings also seems a good strategy. Were women with benign tumors of the ovaries excluded from this study?

In Table 1, is the clinical prognostic factor for which all factors in Cox actually gave a significant result is a multivariate survival analysis?

Does univariate analysis show that TAAR1 expression is an independent prognosis of OS?

Patients’ characteristics are missing.

Author Response

               Munich, 28th July 2021

Dear reviewer,

we thank you for reviewing our manuscript “Trace- amine associated receptor (TAAR1) is a positive prognosticator for epithelial ovarian cancer” and considering it for publication.

In the following section we would like to reply to your queries.

Response letter to Reviewer 1:

In this manuscript, the authors report that high TAAR1 expression may be an indicator of ovarian cancer survival, and show that it is significantly enhanced in low-grade serous carcinoma compared to high-grade serous carcinoma. A positive correlation was also suggested between TAAR1 expression and PR-A and PR-B.

The authors immunohistochemically stained cancer tissues with anti-TAAR1 based on clinical ovarian cancer samples and assessed them by semi-quantitative immune response scores (IRS).

However, this manuscript has weak validation of the role of TAAR1 in ovarian cancer tissue or cancer. Independent verification of this should be provided and these assumptions should be further investigated. In the mechanism part, how TAAR1 contributes to gene expression regulation is completely missing. Therefore, this manuscript is not accepted as is.

The role of TAAR1 in ovarian cancer is yet unknown and independent validation of our study is an important issue. Aim of this study was to identify TAAR1 as a potential prognostic factor and to correlate it with clinical parameters. Since this is a hypothesis generating study, further studies are required to investigate the exact role of TAAR1 and thyroid hormones, respectively, in ovarian cancer. Although we could already investigate TAAR1 signal transduction in breast cancer, it is an open field in ovarian cancer.

Based on clinical ovarian cancer samples, did the authors find a correlation between TAAR1 expression and disease state at the mRNA and protein level? The survival analysis may be studied using both mRNA and protein data such as from IHC data.

Thank you for this interesting remark. In a cell culture model, a correlation between TAAR1 expression and disease state at mRNA and protein level could strengthen our hypothesis and bring further proof. This study, however, is a first step to elucidate a potential role of TAAR1 in ovarian cancer. More studies are required to figure out the role of TAAR1 in ovarian cancer on a molecular level.

Data from public database should be retrieved as well.

We thank you for this comment and added data from the public KM plotter database as shown in section 2.6 and Figure S1 in the supplementary files.

Gene expression analysis requires performing genome-wide analysis of TAAR1-regulated genes to elucidate the most important functions of TAAR1 in ovarian cancer cells.

Thank you for the important remark. mRNA analysis is planned for further studies in ovarian cancer including NGS analysis.

The study included 156 female patients diagnosed with ovarian cancer and who underwent surgery. Four histological subtypes of epithelial ovarian cancer were included in this study. Investigating TAAR1 expression in a relatively large population of patients with serous ovarian cancer and presenting functional findings also seems a good strategy. Were women with benign tumors of the ovaries excluded from this study?

It would be an interesting investigation to additionally evaluate TAAR1 in benign ovarian tumours. This study focused on malign ovarian cancer since the primary focus was on prognostic value of TAAR1 in ovarian cancer. Due to that reason benign tumors were not observed and not included in this study.

In Table 1, is the clinical prognostic factor for which all factors in Cox actually gave a significant result is a multivariate survival analysis?

Yes, Table 1 shows a multivariate Cox regression analysis.

Does univariate analysis show that TAAR1 expression is an independent prognosis of OS?

The univariate analyses showed that TAAR1 is a significant positive prognosticator.

Patients’ characteristics are missing.

Thank you for this comment. Patients’ characteristics were added in a separate table (Table 2).

Yours sincerely,

T.L.R. Vogelsang, A. Vattai, U. Jeschke and F. Trillsch

Reviewer 2 Report

This study was to investigate the prognostic value and the correlation of TAAR1 with clinical parameters in epithelial ovarian cancer. I had some suggestions.

  1. The clinical information of patients enrolled in the study seemed to be inadequate. The well-known clinical prognostic factors for epithelial ovarian cancer includes histological subtypes, tumor grade, FIGO stage (or TMN stage), residual tumor volume of cytoreduction surgery and adjuvant chemotherapy. I suggest the authors to provide a Table for these clinical parameters in detail.
  2. It was redundant to present the IHC from Figure 1 to 5. Please revise and provide it in a concise manner.
  3. It was confused that which TAAR1 expression (membrane or cytoplasm) was selected as the prognostic factor in the study. Please clearly define it.
  4. The Cox regression analysis for overall survival should include the possible clinical prognostic factors, including histological subtypes, tumor grade, FIGO stage (or TMN stage), residual tumor volume of cytoreduction surgery and adjuvant chemotherapy, in addition to TAAR1 expression.
  5. What is the rationale to correlate TAAR1 and other variables (estrogen receptor- α, progesterone receptor- A, progesterone receptor- B, nuclear vitamin D receptor, G-protein-167 coupled estrogen receptor/G-protein-coupled receptor 30 (GPER/GPR30), Muc-115D8 169, VU3C6 and Glycodelin A) in result section 2.5 and Table S1?
  6. The patients between 1990 and 2002 were included in this study. The local ethics committee approved this study (227-09) on 30th September 2009. How were these Informed consents obtained from all the subjects involved in the study prior to study participation?

Author Response

               Munich, 28th July 2021

Dear reviewer,

we thank you for reviewing our manuscript “Trace- amine associated receptor (TAAR1) is a positive prognosticator for epithelial ovarian cancer” and considering it for publication.

In the following section we would like to reply to your queries.

Response letter to Reviewer 2:

This study was to investigate the prognostic value and the correlation of TAAR1 with clinical parameters in epithelial ovarian cancer. I had some suggestions.

  1. The clinical information of patients enrolled in the study seemed to be inadequate. The well-known clinical prognostic factors for epithelial ovarian cancer includes histological subtypes, tumor grade, FIGO stage (or TMN stage), residual tumor volume of cytoreduction surgery and adjuvant chemotherapy. I suggest the authors to provide a Table for these clinical parameters in detail.

We thank you for this important objection. We now included a Table with all the clinical parameters (Table 2). As the collective has been collected from 1992 until 2002, the residual tumor volume of cytoreduction surgery was not evaluated at that time and adjuvant chemotherapy was not filed at that time.

  1. It was redundant to present the IHC from Figure 1 to 5. Please revise and provide it in a concise manner.

We revised Figure 4 and 5 and assembled them into one figure (Figure 4). Since this study had a strong focus on immunohistochemistry, we found it important to provide decent footage of histopathological slices.

  1. It was confused that which TAAR1 expression (membrane or cytoplasm) was selected as the prognostic factor in the study. Please clearly define it.

TAAR1 can be expressed in both cytoplasm and membrane of ovarian cancer tissue. For the analysis regarding overall survival, the general expression in cytoplasm and/or membrane had been looked at, meaning if there was any expression of TAAR1 at all (as described in section 2.4). Other analyses had been carried out with either cytoplasmic or membrane expression as described in the respective parts of the manuscript.

  1. The Cox regression analysis for overall survival should include the possible clinical prognostic factors, including histological subtypes, tumor grade, FIGO stage (or TMN stage), residual tumor volume of cytoreduction surgery and adjuvant chemotherapy, in addition to TAAR1 expression.

We thank you for this specification and now included the possible clinical factors age, histological subtype, grading and FIGO stage to the Cox regression analysis (Table 1). Residual tumor volume of cytoreductive surgery and adjuvant chemotherapy had not been collected for this patients’ cohort.

  1. What is the rationale to correlate TAAR1 and other variables (estrogen receptor- α, progesterone receptor- A, progesterone receptor- B, nuclear vitamin D receptor, G-protein-167 coupled estrogen receptor/G-protein-coupled receptor 30 (GPER/GPR30), Muc-115D8 169, VU3C6 and Glycodelin A) in result section 2.5 and Table S1?

Thank you for this question. These parameters have been evaluated in this patients’ cohort in previous studies. Therefore, we were able to compare the TAAR1 expression with already published results of our group on the same collection of patients. To elucidate a potential link between TAAR1 and these variables we had performed correlation analyses and included those in this study.

  1. The patients between 1990 and 2002 were included in this study. The local ethics committee approved this study (227-09) on 30th September 2009. How were these Informed consents obtained from all the subjects involved in the study prior to study participation?

This is a valuable objection, thank you. We have taken this passage out of the paper since no informed consent was needed as per declaration of our ethics committee because the material used was left-over material and all diagnostic procedures had been carried out. Please excuse this confusion.

Yours sincerely,

T.L.R. Vogelsang, A. Vattai, U. Jeschke and F. Trillsch

Round 2

Reviewer 1 Report

The revised version apparently addressed all of my comments and performed additional data.

These data greatly improve the quality of the paper and the authors conclusions are now justified by experimentation.

Reviewer 2 Report

It is acceptable for publication.